

# Evaluating the role of post-harvest glyphosate application in enhancing weed control in winter wheat

Ahmet Tansel Serim[1], Ünal Asav[2], Yalçın Kaya[3], Bülent Başaran[3] and Eric L. Patterson[4]

[1] Department of Plant Protection, Bilecik Seyh Edebali University, Bilecik, Turkey
[2] Department of Plant Protection, Gaziosmanpasa University, Tokat, Turkey
[3] Middle Black Sea Transitional Zone Agricultural Research Institute, Tokat, Turkey
[4] Department of Plant, Soil and Microbial Sciences, Michigan State University, East Lansing, MI, United States of America

Corresponding author
Ahmet Tansel Serim,
ahmettansel.serim@bilecik.edu.tr

## ABSTRACT

Changes in the timing and intensity of spring rainfall have led to a significant increase in late-season weed emergence in Türkiye. These newly emerged weeds tend to grow more vigorously due to the absence of competition with crops and other weeds during their development. Two field experiments were conducted in continuous monoculture winter wheat over three growing seasons (2020–2023) in Türkiye. The first goal was to determine the impact of post-harvest herbicide (PHH) on the critical time for weed removal (CTWR) in winter wheat, and the second goal was to evaluate the effects of PHH combined with various weed control treatments on weed populations, the soil seed bank, and crop yield. The experiment followed a split-plot design, with the PHH regimes and weed removal timing or weed control treatments serving as the main and sub-plots, respectively. The herbicide regime included post-harvest glyphosate potassium salt (PHG) applied at 2.646 kg ai ha$^{-1}$ and No PHG. Weed removal timings were set at 10-day intervals, from 0 to 110 days after wheat emergence (DAE). Weedy and weed-free controls were included for comparison. The weed control treatments involved post-emergence tribenuron-methyl at 7.5 g ai ha$^{-1}$ and hand weeding. The application of the PHG delayed the CTWR from 416 growing degree days (GDD) to 516.5 GDD in 2022 and from 465.6 GDD to 661.2 GDD in 2023, effectively preventing yield loss. The combined use of PHG with post-emergence tribenuron-methyl or hand weeding maximized wheat yield while minimizing the weed flora and the size of the soil seed bank.

## INTRODUCTION

Wheat is a valuable staple crop for humanity as it provides the main dietary energy and protein for one in five people in the world (*Erenstein et al., 2022*). Its prominence is rooted in several reasons, such as wheat's adaptability to diverse climates, easy husbandry practices, and flexibility to adverse conditions. Contrary to these advantage over other crops, extreme weather events, pests, diseases, and weeds often prevent high grain yield (*Mao et al., 2023*).

Weeds are generally considered strong competitors to wheat, and they result in severe yield losses depending on weed species, density, and competition timing (*Peairs, Bean & Gossen, 2005*). To mitigate these undesirable impacts, many weed control treatments, including biological, cultural, mechanical, physical, and chemical techniques are commonly used by farmers.

To address these challenges, cultural weed control, such as cultivar selection, changing seeding rate, and crop rotation, has been successfully employed in many fields (*Beres et al., 2010*; *Peairs, Bean & Gossen, 2005*), but the widespread use of this technique is often constrained by low rainfall or the inability to provide workers. Chemical weed control by herbicides, on the other hand, may allow growers to control them easily and cost-effectively (*Coleman et al., 2019*); therefore, making herbicides indispensable tools for many growers. Herbicide use has extremely popular among wheat growers until the last decades (*Coleman et al., 2019*; *Peairs, Bean & Gossen, 2005*).

The incessant use of herbicides has led to several adverse side effects, such as carryover, runoff, drift, biodiversity loss, and herbicide-resistant weeds, especially in many wheat-producing regions (*Wang et al., 2022*; *Türkseven et al., 2022*). For example, there have been 533 documented cases of herbicide-resistant weeds worldwide, more than half of which are in wheat fields (*Heap, 2024*). Resistance is also becoming widespread in Turkish wheat fields, with five weed species reported to be resistant to ALS (Group 2) and/or ACCase (Group 1) inhibitor herbicides (*Heap, 2024*). Herbicide-resistant weeds and the side effects of wheat herbicides threaten the sustainability of wheat production and therefore force researchers to find new alternatives. Growers often apply herbicides in a mixture, overrate, or repeat them to control herbicide-resistant weeds, but this approach raises the cost. At this point, pre-sowing (early season) and post-harvest (late season) herbicide applications might increase weed control efficacy while decreasing risks related to the appearance of new herbicide-resistant weeds in crops. The first is mainly used to burndown weeds before the crop planting or reduce tillage requirements. While, the second method is commonly used to control late-season weeds and prevent the production of their seeds for the next season.

The spring rainfall patterns in Anatolia, Türkiye have been changing in recent years. For instance, the cumulative rainfall in June increased by 78% during the experiment compared with the long-term records of Kazova, Tokat, Türkiye (Table 1). This change stimulates the emergence of some weeds during late spring and allows them to escape in-season herbicides. Moreover, some of late emerged weeds prevent harvest, especially in continuous monoculture wheat fields because they cannot complete their life cycle when winter wheat reaches harvest time (Fig. 1A). Farmers generally abandon patches that are covered by weeds in winter wheat fields and reluctantly allow the weeds to produce their seeds (Fig. 1B). Post-harvest weed control methods, such as tillage or herbicides (both contact and systemic) are used to prevent adverse impacts of these late-season weeds. Post-harvest glyphosate application has been shown that to suppress the growth of late-season weeds and reduce seed production (*Crow et al., 2015*). Similarly, *Antier et al. (2020)* reported that pre-sowing, pre-emergence, and post-harvest glyphosate applications are commonly used to control weeds and volunteers in annual cropping systems, such as

**Table 1 Climate data of experimental field from 2020–2023 and long-term (*Turkish State Meteorological Service (MGM), 2024*).**

| Month | Average temperature (°C) | | | | Cumulative rainfall (mm) | | | | Long-term (1929–2023) | |
|---|---|---|---|---|---|---|---|---|---|---|
| | 2020 | 2021 | 2022 | 2023 | 2020 | 2021 | 2022 | 2023 | Temperature (°C) | Cumulative rainfall (mm) |
| January | 3.0 | 5.0 | 1 | 2.6 | 52.8 | 62.7 | 43.3 | 5.8 | 1.9 | 40.9 |
| February | 4.0 | 5.2 | 4.7 | 2.2 | 66.2 | 8.7 | 25.9 | 32.8 | 3.5 | 33.8 |
| March | 9.8 | 6.0 | 3 | 9.0 | 32.9 | 71.2 | 48.9 | 35.0 | 7.4 | 40.8 |
| April | 11.1 | 13.7 | 14.5 | 12.2 | 22.0 | 14.2 | 33.2 | 114.5 | 12.5 | 54.2 |
| May | 17.0 | 18.5 | 15.2 | 15.4 | 35.9 | 54.6 | 32.6 | 61.0 | 16.5 | 58.9 |
| June | 20.7 | 20.0 | 20.7 | 20.0 | 81.6 | 55.5 | 55.1 | 71.8 | 19.9 | 38.2 |
| July | 24.2 | 24.1 | 20.7 | 21.7 | 1.4 | 27.7 | – | 35.6 | 22.3 | 11.2 |
| August | 22.3 | 23.6 | 24.9 | 24.0 | 1.1 | 17.9 | 4.7 | 4.3 | 22.4 | 5.6 |
| September | 23.0 | 17.8 | 19.4 | 18.9 | 1.0 | 27.1 | 27.4 | 14.3 | 18.8 | 17.7 |
| October | 18.0 | 13.0 | 13.3 | 13.7 | 0.1 | 10.5 | 35.3 | 4.1 | 13.7 | 39.3 |
| November | 7.4 | 15.2 | 9.3 | – | 11.1 | 0.1 | 31.3 | – | 7.9 | 44.0 |
| December | 5.5 | −0.1 | 6.1 | – | 17.2 | 13.0 | 17.5 | – | 3.8 | 47.1 |

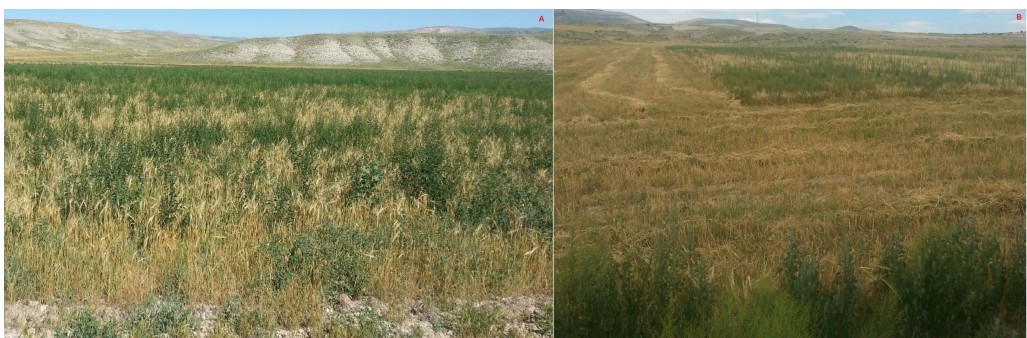

**Figure 1 Winter wheat field heavily infested with weeds at harvest time (A); the patch abandoned due to weed infestation (B).**

cereals, maize, legumes, and sugar beets, in many European countries. Although glyphosate has been registered for post-harvest stubble treatment in some countries, no commercial product has been registered in Türkiye for this purpose.

Because of its protective structure, soil acts as a natural vault for weed seeds and influences the makeup of weeds in succeeding crops. The weed seed bank is affected by several factors, such as crop rotation, soil characteristics, and weed management techniques (*Jabran & Chauhan, 2015*; *Skuodienė, Matyžiūtė & Šiaudinis, 2024*; *Osipitan et al., 2018*; *Schwartz-Lazaro & Copes, 2019*). For instance, *Mickelson et al. (2004)* indicated that post-harvest glyphosate application reduced Kochia seed production by up to 99%. However, the impact of post-harvest glyphosate, alone or in combination with other

post-emergence herbicides, on the weed seed bank in wheat production has not been thoroughly investigated in the literature.

The critical time for weed removal (CTWR) is the beginning of the critical period of weed control, which is a useful tool in integrated weed management strategies. CTWR provides valuable knowledge to growers concerning when weed control efforts should be implemented to prevent yield losses. Indeed, previous studies have shown that CTWR in wheat typically spans the first 3 weeks after crop emergence (*Agostinetto et al., 2008*). Pre-sowing glyphosate application to no-till maize effectively eliminates weeds and postpones CTWR for up to 17 days (*Adamič Zamljen & Leskovšek, 2024*). Similar findings were also reported by *Roncatto et al. (2023)* in soybean, *Ulusoy et al. (2021)* in corn, *Knezevic et al. (2013)* in sunflower, and *Barnes et al. (2019)* in popcorn. Although post-harvest herbicide applications control weed and weed seed production in stubble, their impact on the CTWR has not been studied.

A comprehensive weed management approach, such as involving post-harvest glyphosate applications, followed by post-emergence herbicide or hand weeding, may be necessary to effectively manage the dynamic weed flora in wheat production systems. This study aimed to determine the effectiveness of post-harvest glyphosate potassium salt application combined with or without the weed control methods, such as post-emergence herbicide and hand weeding, on weeds, wheat grain yield, the weed seed bank, and the CTWR in non-herbicide resistant winter wheat.

## MATERIALS AND METHODS

### Experimental field and environmental conditions

Field experiments were conducted at the Middle Black Sea Transaction Zone Agricultural Research Institute (40.1928 N; 36.2656E), Kazova, Tokat, Türkiye, from 2020 to 2023. The soil was clay loam with 1.5% organic matter and a pH of 7.9. The Flamura-85 (Flamura-85®; TAREKS, Balıkesir, Türkiye) winter wheat variety (*Triticum aestivum* L. var. Flamura-85) was sown at a seeding rate of 500 seeds $m^{-2}$ during October-November. This variety is widely cultivated in the central Anatolian region and is known to resist cold and lodging.

Kazova has a continental climate with distinct seasonal variations. The coldest month is January at $-1.6 \,°C$, whereas the warmest month is August at $29.9 \,°C$. The first frost was typically recorded in October or November, and the last frost was recorded in May. The annual average number of rainy days was 103.8, with a value of 431.7 mm. April, May, and June were generally the wettest months, whereas the lowest rainfall was recorded in July and August. The specific climate data collected during the experiment are shown in Table 1.

### Research design and treatments
#### Impact of herbicide programs on weed flora and wheat yield
The study was designed as a split-plot experiment with four replications. The main plots were assigned to the herbicide regime post-harvest glyphosate potassium salt (PHG) treatment and No PHG, and the subplots were allocated to the weed control treatments.

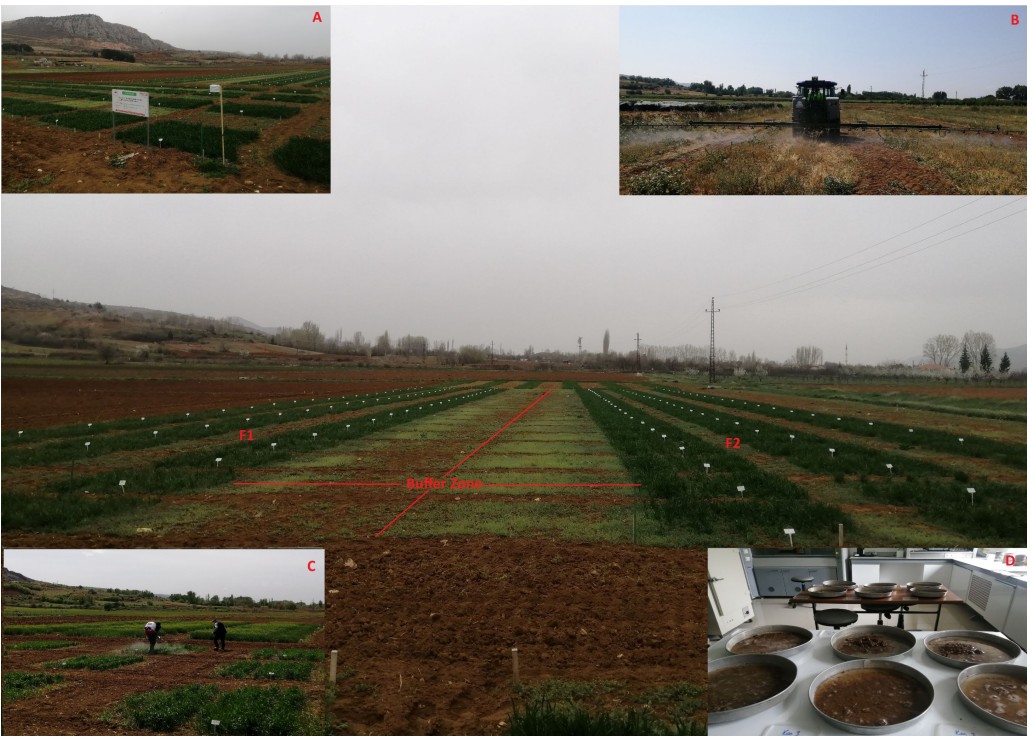

**Figure 2 Experimental area, parcel locations, and treatments at Kazova.** (A) Experimental field, (B) post-harvest glyphosate treatment (PHG), (C) post-emergence tribenuron-methyl application, (D) separation of weed seeds from soil taken from control plots (T4: No PHG + Weedy check), F1: PHG treated blocks, F2: No PHG blocks.

Each plot was 3 m × 3.5 m. A two m alley was left between blocks, and a one m alley was left between parcels (Fig. 2A). The experiment was conducted in a field with winter wheat grown for 3 consecutive years. Four of the eight blocks were allocated for PHG application, whereas the remaining four were used for No PHG (conventional method) application. These blocks were marked, and the same treatments were repeated for 3 years. The same plot was used for each treatment when the experiment was repeated in the second and third years in the winter wheat.

The treatments used were as follows: T1: PHG + Post-emergence tribenuron-methyl, T2: PHG + Weedy check, T3: No PHG + Post-emergence tribenuron-methyl, T4: No PHG + Weedy check (Control), T5: PHG + Hand weeding, and T6: No PHG + Hand weeding. Hand weeding was performed 3 times at 10 day intervals after tillering began. Although this treatment is a time-consuming and labor-intensive method, it is advised by local advisers and is practiced by Turkish growers whose fields are covered by volunteers and weeds that have not been controlled by herbicides, such as feral rye or herbicide-resistant weeds.

Glyphosate potassium salt (ROUNDUP STAR®; Bayer Crop Science, İstanbul, Türkiye) and tribenuron-methyl (GRANSTAR®; FMC Turkey, İstanbul, Türkiye) were applied using a field sprayer and motorized backpack sprayer adjusted to deliver 200 l ha$^{-1}$ at 200 kPa pressure, respectively (Figs. 2B and 2C). Glyphosate potassium salt was applied at

2.646 kg ai ha$^{-1}$ one week after wheat harvest as a post-harvest treatment to control grass and broadleaf weeds in the stubble. Tribenuron-methyl was applied at a rate of 7.5 g ai ha$^{-1}$ at the 3–4 leaf stage of broadleaf weeds. Glyphosate potassium salt has no residual activity, whereas tribenuron-methyl has a short residual activity in the soil (*Shaner, 2014*).

The weed flora was determined at the end of April 3 weeks after tribenuron-methyl treatment. Weeds in each plot were recorded from one m$^2$ and identified according to the Flora of Turkey and the North Aegean Islands (*Davis, 1965–1985*). During the harvest time, spikes in a one m$^2$ area were harvested from each plot. The harvested spikes were placed in bags, brought to the laboratory, and threshed. The grain yield was calculated on a per-hectare basis by converting the results from the harvest area. Weed density was calculated by using the Formula (1) (*Nkoa, Owen & Swanton, 2015*).

$$D = \frac{\sum Y}{Sa} \tag{1}$$

where D is the density of the weed species; $\sum Y$ is the count of individual plants of weed species placed in a quadrat; and Sa is the surface area of the quadrat (one m$^2$).

The collected data on wheat grain yield were subjected to variance analysis using the Agricolae package (*Mendiburu & Yaseen, 2020*) in R statistical software (*R Studio Team, 2024*). The means were compared using the Least Significant Difference test at the 5% probability level. The wheat yield data were subjected to ANOVA separately for each year because the post-harvest herbicide treatment by weed control treatment interactions were significantly different ($P < 0.05$).

### Impact of herbicide programs on the weed seed bank

The weed seed bank was determined for three consecutive years by using the method described by *Mayor & Dessaint (1998)*, with slight modifications. Soil samples were collected from each plot using a soil borer (five cm diameter) at a 10 cm soil depth, and each soil sample was nearly 500 g. The soil samples were first sieved (four mm × four mm) to remove unwanted materials, such as debris and soil particles, and then sieved using a precise sieve (0.25 mm × 0.25 mm) to separate the soil aggregates. The soil samples were placed in aluminum trays filled with tap water and left for 24 h (Fig. 2D). The slurry was gently mixed and washed using a sieve set under tap water. The seeds were dried using towel papers, put in paper bags, and stored in a cooler at +4 °C until classification. Weed species in the seed bank were identified using the Flora of Turkey and the North Aegean Islands (*Davis, 1965–1985*).

The impacts of the PHG and weed control treatments on Average Weed Seed Density (AWSD) data were subjected to variance analysis using the Agricolae package (*Mendiburu & Yaseen, 2020*) in R statistical software (*R Studio Team, 2024*). The means were compared using the Least Significant Difference test at the 5% probability level to determine the statistical significance of the differences among the treatments. AWSD data were subjected to ANOVA separately for each year because year, PHG treatment by weed control treatment interactions were significantly different ($P < 0.05$).

*Impact of the PHG on the critical time for weed control (CTWR)*

The CTWR in wheat was calculated by assessing the relationship between wheat grain yield and the duration of weed presence during the growing season. In the weedy control treatment, weeds were allowed to compete with the crop for the entire season, whereas in the weed-free control, weeds were removed manually at 10-day intervals to prevent competition with the crop. Weed removal times were adjusted at 10-day intervals, from 0 to 110 days after wheat emergence (DAE). The grain yield obtained from the weed-free control treatment was considered the maximum yield, and the percentage yield loss was calculated by comparing the grain yield from the weedy control treatment to that of the weed-free control.

The experiment was carried out for three consecutive years in a field grown continuous winter wheat. The experimental design was a split-block design with three replications, where the main plots were assigned to the PHG and No PHG, and the subplots were allocated to the length of the weed-free period, weed-free, and weedy season-long controls. Three of the six blocks were allocated for PHG application, whereas the remaining three were used for No PHG (conventional method) application. These blocks were marked, and the same application was repeated for 3 years. Each plot was two m × three m. A non-linear regression analysis was performed on the yield loss data to model the relationship between weed presence duration and wheat grain yield (*Knezevic & Datta, 2015*). The log–logistic model was then used to estimate the CTWR (Formula 2).

$$Y = c + \frac{d - c}{1 + \exp(b(\log(x) - \log(E)))} \tag{2}$$

where $Y$ is the grain yield; $c$ is the lower limit; $d$ is the upper limit; $x$ is the weed removal timing expressed in growing degree days (GDD) after wheat emergence; $E$ is the GDD at the inflection point ($I_{50}$); and $b$ is the slope around the $I_{50}$. The GDD was calculated according to Formula (3) (*Mcmaster & Wilhelm, 1997*):

$$GDD = \sum \left[ \frac{T_{max} + T_{min}}{2} \right] - T_{base} \tag{3}$$

where GDD, $T_{max}$, $T_{min}$, and $T_{base}$ were growing degree days, daily maximum temperature, daily minimum temperature, and base temperature ($\geq 0\,°C$) for wheat growth, respectively. The data are presented separately for each year because the year and herbicide treatment interaction were significant ($P < 0.05$).

## RESULTS AND DISCUSSION

### Impact of herbicide programs on weed flora and wheat yield

The experimental field consisted of 15 weed species, most of which were annual dicotyledonous (Table 2). The most prevalent weeds in the experimental field were *Veronica hederifolia* L., *Chenopodium album* L., and *Amaranthus retroflexus* L. The composition and density of weed species in the herbicide-treated and untreated plots varied over the field trials.

Post-emergence tribenuron-methyl controlled some weeds such as *C. album*, *C. arvense*, *F. officinalis*, *S. arvensis*, *P. oleracea*, *L. serriola*, *S. nigrum*, and *A. retroflexus*, whereas it

**Table 2** Impact of post-harvest glyphosate (PHG) with and without post emergence tribenuron-methyl applied on weed density during 2021–2023 (Plant m$^{-2}$).

| Weed species | PHG+Post-em Tribenuron-methyl | | | PHG+weedy control | | | No PHG+Post-em Tribenuron-methyl | | | No PHG+weedy control | | |
|---|---|---|---|---|---|---|---|---|---|---|---|---|
| | 2021 | 2022 | 2023 | 2021 | 2022 | 2023 | 2021 | 2022 | 2023 | 2021 | 2022 | 2023 |
| VERHE | 7.0 | 2.5 | 1.3 | 11.5 | 9.5 | 10.3 | 15.0 | 12.3 | 15.0 | 12.5 | 8.5 | 14.3 |
| XANST | 2.0 | 3.5 | 1.5 | 0.5 | 2.5 | 0.5 | 2.5 | 3.5 | 3.5 | 1.0 | 2.00 | 3.8 |
| CIRAR | – | – | – | 2.0 | 3.0 | 3.0 | 1.5 | 0.3 | 0.5 | 5.5 | 3.3 | 2.8 |
| CONAR | 4.0 | 3.0 | 1.5 | 4.3 | 1.5 | 2.8 | 6.3 | 7.5 | 5.0 | 5.5 | 6.0 | 7.3 |
| CHEAL | 16.0 | 15.5 | 7.3 | 12.0 | 13.8 | 14.0 | 8.0 | 11.3 | 8.8 | 8.3 | 6.5 | 4.5 |
| FUMOF | – | – | – | 1.5 | 2.8 | 6.8 | 3.0 | 2.0 | 1.5 | 1.0 | 4.0 | 3.8 |
| GALAP | – | – | – | 3.5 | 2.8 | 7.5 | 1.0 | – | – | – | 2.0 | 3.0 |
| SINAR | – | – | – | – | 1.5 | 2.5 | 1.5 | 0.3 | 1.3 | 2.5 | 3.0 | 4.3 |
| POLAV | – | – | – | – | 1.0 | 2.0 | – | – | – | – | 2.0 | 3.0 |
| POROL | – | – | – | – | – | – | – | 1.5 | 1.00 | – | 2.3 | 3.0 |
| LACSE | – | – | – | – | 2.3 | 3.5 | – | – | 0.50 | – | 1.0 | 3.0 |
| SOLNI | – | – | – | – | – | – | – | – | – | 0.5 | – | – |
| SETVI | 1.5 | – | – | 0.5 | 1.0 | 3.8 | – | 1.0 | 0.8 | 3.0 | 3.0 | 1.5 |
| AMARE | 6.5 | 2.5 | 2.5 | 4.0 | 5.0 | 6.8 | 10.3 | 11.5 | 14.3 | 8.0 | 9.0 | 10.5 |
| ALOMY | – | 7.5 | 1.8 | – | 3.0 | 2.8 | – | 1.25 | 2.0 | – | 4.3 | 3.0 |
| Total | 37.0 | 34.5 | 15.9 | 39.8 | 49.7 | 66.3 | 49.1 | 52.5 | 54.2 | 47.8 | 56.9 | 67.8 |
| Species number | 6 | 6 | 6 | 9 | 13 | 13 | 9 | 11 | 12 | 11 | 14 | 14 |
| Average Weed Density | 6.2 | 5.8 | 2.7 | 4.4 | 3.8 | 5.1 | 5.5 | 4.8 | 4.5 | 4.8 | 4.1 | 4.8 |

**Notes.**

VERHE, *Veronica hederifola* L.; XANST, *Xanthium strumarium* L.; CIRAR, *Cirsium arvense ( L.) Scop.*; CONAR, *Convolvulus arvensis* L.; CHEAL, *Chenopodium album* L.; FUMOF, *Fumaria officinalis* L.; GALAP, *Galium aparine* L.; SINAR, *Sinapis arvensis* L.; POLAV, *Polygonum aviculare* L.; POROL, *Portulaca oleracea* L.; LACSE, *Lactuca serriola* L.; SOLNI, *Solanum nigrum* L.; SETVI, *Seteria viridis* L.; AMARE, *Amaranthus retroflexus* L; ALOMY, *Alopecurus myosoroides* Huds.

partly controlled *X. strumarium*, *P. aviculare*, and *C. arvensis*. However, newly emerged seedlings were observed 3 weeks after treatment because of the low soil persistence of the herbicide (data not shown). Post-emergence tribenuron-methyl has no effect on *V. hederifolia*, *G. aparine*, *S. viridis*, or *A. myosuroides.* However, PHG application alone resulted in considerable control of these weeds in stubble.

The average weed density (AWD) in No PHD + weedy check plots, which also meant naturally occurring weed flora, remained relatively stable over 3 years, but the weed count steadily increased and reached 141.8% of that in the first year in the third year (Table 2). In parcels applied with post-emergence tribenuron-methyl, AWD slightly decreased over time, whereas weed count slightly increased. PHG + weedy check application had effects similar to those of post-emergence tribenuron-methyl application in the first two years. However, AWD and weed count increased in the third year. This unexpected increase might be attributed to the heavy rainfall that occurred in April 2023 (Table 1). The most significant reduction in AWD and weed count was observed in parcels treated with a PHG + post-emergence tribenuron-methyl. The AWD and weed count decreased from 6.2 plant m$^{-2}$ and 37 plants to 2.7 plant m$^{-2}$ and 15.9 plants, respectively. Another aspect is the impact of these treatments on weed species abundance. Neither PHG nor post-emergence

tribenuron-methyl treatments caused a negative impact on the increase of weed species abundance in a short timescale, but PHG + post-emergence tribenuron-methyl did.

Previous studies have shown that weed control treatments such as tillage or herbicides might alter weed flora in agricultural fields (*Tørresen et al., 2003*; *Shahzad et al., 2016*; *Ball & Miller, 1993*). The strong impact of herbicides on weed density and diversity in wheat fields was reported by *Culpepper (2006)*, *Shahzad et al. (2021)*, *Wilson et al. (2007)*, and *Barnes et al. (2019)*. *Young & Thorne (2004)* stated that the use of late-season herbicides in no-till caused a significant weed shift, and *Salsola iberica* became the prevalent weed species in the field. Contrary, *Reddy, Bryson & Nandula (2015)* showed that there was no meaningful impact of post-harvest herbicide treatment (pendimethalin+paraquat) on the weeds compared with no herbicide treatment. This discrepancy might be caused by herbicides used or specific weed species in the flora. Specifically, PHG application was particularly effective at reducing the density of VERHE and AMARE, both of which were less prevalent after herbicide treatment. This application effectively suppressed the dominance of these weeds and provided favorable conditions for the proliferation of CHEAL, as the reduced competition allowed CHEAL to become the primary weed, similar to the results of *Wilson et al. (2007)*.

The PHG + hand weeding (T5) treatment resulted in the highest wheat grain yield of $5.2-7.5$ t ha$^{-1}$, which was 37–41% greater than the No PHG + weedy check (T4) treatment (Fig. 3). In comparison, PHG + tribenuron-methyl treatment was as effective as PHG + hand weeding application, with a $5.1-7.2$ t ha$^{-1}$ grain yield. Furthermore, No PHG + hand weeding resulted in a $1.28-1.8$ t ha$^{-1}$ increase in grain yield compared to No PHG + weedy check treatment. However, No PHG + tribenuron-methyl or PHG + weedy check treatments provided a 17–25% lower wheat yield than the PHG + hand-weeding treatment. Altogether, these results indicated that the use of PHG was effective as a post-emergence herbicide to control weeds in winter wheat.

Across all the treatments, the grain yield tends to increase over the three years (2021 to 2023), suggesting that most of the treatments have a positive impact, even though the differences between them are not statistically significant. Moreover, the grain yield in all treatments in the third year was greater than that in the previous years because of the greater rainfall in that year (Table 1), which is in line with the results of the study conducted by *Gandía et al. (2021)* who reported that the most powerful impact on yield parameters and weed populations was rainfall during the growing season.

There is a strong negative relationship between weed abundance and crop yield in almost all cropping systems, whereby an increase in weed density directly impacts the productivity of crops. Weed-crop competition can significantly hinder crop development, leading to lower yields. Consistent with these findings, former studies have indicated that PHG is a powerful tool for controlling many problematic weeds in crops during the late season (*Young & Gealy, 1986*; *Kumar & Jha, 2015*). For example, *Kumar et al. (2021)* indicated that PHG application at 1,260 g ae ha$^{-1}$ controlled palmer amaranth at 93% 8 weeks after treatment. Similarly, *Young & Gealy (1986)* found that post-harvest weed control not only reduces weed seeds and aboveground biomass but also saves soil moisture for subsequent crops. This finding particularly explained the yield increase in the plots treated with PHG.

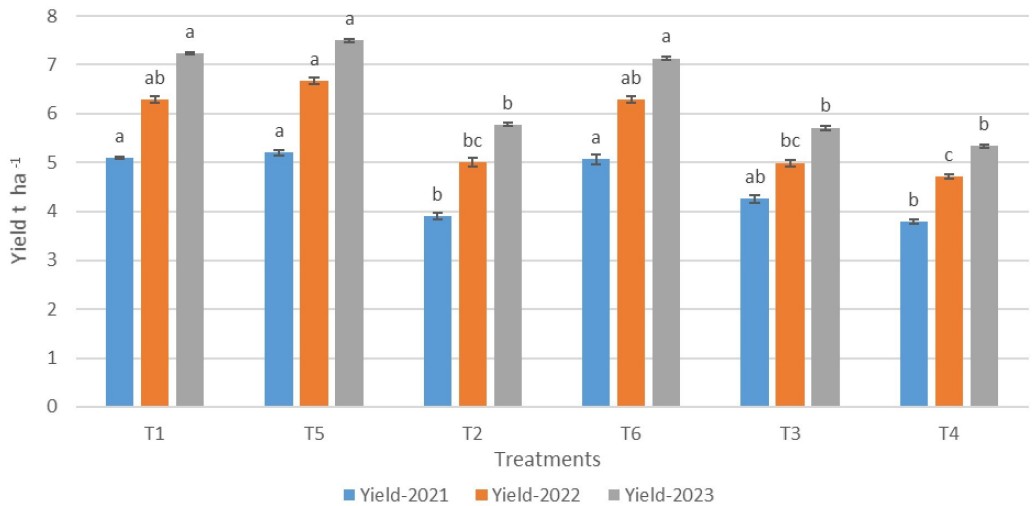

**Figure 3** **Wheat yield response to treatments from 2021 to 2023.** (T1: Post-harvest glyphosate (PHG)+Post-em Tribenuron; T2: PHG+Weedy check; T3: No PHG+Post-em tribenuron; T4: No PHG+Weedy check, T5: PHG+Hand weeding, T6: No: PHG+Hand weeding). Lowercase letters represent statistical significance, where different letters indicate significant differences between treatments in the same year ($P < 0.05$).

Even though post-harvest glyphosate potassium salt application suppressed weed competition and increased wheat yield, this impact was limited because of the non-residual efficacy of glyphosate potassium salt. Maintaining weed control efficacy can be possible through herbicide application programs such as post-harvest glyphosate potassium salt followed by selective herbicides or tillage. Consistent with our findings, *Bae et al. (2022)* reported that post-harvest herbicide (dichlobenil) along with post-emergence herbicides (clopyralid/mesotrione followed by mesotrione + sethoxydim) on *Equisetum arvense* L. was 47–52% more effective than only post-emergence herbicide (sulfentrazone followed by clopyralid/mesotrione followed by mesotrione + sethoxydim). A similar study by *Meyers et al. (2016)* supports our findings, demonstrating that an herbicide program incorporating post-harvest application of flumioxazin achieved greater control of problematic weeds than did a program without post-harvest treatment.

## Impact of herbicide programs on weed seed bank

The soil seed bank consisted of 19 weed species (Supplemental Information). The weed seeds in the experimental field were *Amaranthus retroflexus* L., *Veronica hederifola* L., *Portulaca oleracea* L., *Chenopodium album* L., *Seteria viridis* L., *Solanum nigrum* L., *Stellaria media* (L.) Vill., *Poligonum aviculare* L., *Consolida regalis* Gray, *Xanthium strumarium* L., *Euphorbia* spp., *Alopecurus myosoroides* Huds., *Cirsium arvense* ( L.) Scop., *Convolvulus arvensis* L., *Fumaria officinalis* L., *Galium aparine* L., *Sinapis arvensis* L., *Seteria italicus* L., and *Lactuca serriola* L. The species composition and their density in the treated plots varied over the field trials compared to non-treated checks.

**Table 3** Average weed seed density (seeds soil sample$^{-1}$) at 10 cm soil depth in herbicide-treated and control plots before sowing winter wheat.

| Treatment | 2021 | 2022 | 2023 |
|---|---|---|---|
| PHG+post-em tribenuron-methyl | 9.8 (2.1) a | 12.9 (0.5) ab | 5.3 (1.2) b |
| PHG+weedy check | 5.4 (0.8) a | 7.8 (2.1) b | 14.9 (2.3) ab |
| PHG+hand weeding | 13.7 (3.3) a | 8.8 (2.8) b | 7.7 (1.4) b |
| No PHG+Post-em tribenuron-methyl | 13.3 (2.7) a | 17.4 (2.0) a | 22.2 (3.9) a |
| No PHG+weedy check | 8.2 (2.4) a | 18.4 (2.0) a | 24 (8.1) a |
| No PHG+hand weeding | 12.5 (1.3) a | 17 (2.3) a | 24.7 (4.5) a |

**Notes.**
Values in parentheses indicate the standard error of the means; PHG, Post-harvest glyphosate.
Lowercase letters next to values represent statistical significance, where different letters indicate significant differences between treatments in the same year ($P < 0.05$).

The PHG + post-emergence tribenuron-methyl and PHG + hand weeding treatments were the most effective herbicide treatments to reduce AWSD (Table 3). PHG + post-emergence tribenuron-methyl decreased the AWSD by 77.9%, indicating highly effective long-term weed suppression. This meant a considerable reduction compared to the result in 2021 (9.8 seeds soil sample$^{-1}$). Similarly, PHG + hand weeding resulted in a significant decline in AWSD from 13.7 seeds soil sample$^{-1}$ in 2021 to 7.7 seeds soil sample$^{-1}$ in 2023. This decline may be caused by the complementary effect of the PHG and post-emergence weed control treatments (tribenuron-methyl or hand weeding) due to preventing the life cycle of weeds. However, PHG, hand weeding, or tribenuron-methyl applications did not prevent AWSD increase and the replenishment of the soil seedbank throughout the experiment. This was an expected result since none of these treatments had a residual impact to cover a season.

*Kumar et al. (2021)* showed that late-season total herbicide treatments such as glyphosate, dicamba, paraquat, and glufosinate reduced palmer amaranth seed production by more than 90%. Similarly, in another study, *Mickelson et al. (2004)* reported that PHG application at 631 g ha$^{-1}$ resulted in a reduction in Kochia seed production by up to 99%. Furthermore, *Kumar & Jha (2015)* reported that post-harvest herbicide application could reduce seed production of *Kochia scoparia* by 32–100% depending on the herbicide and the combinations used. Additionally, *Maity et al. (2022)* also showed that late-season weed seed control tactics for 4 years resulted in a decrease in *Lolium perenne* ssp. *multiflorum* seeds, similar to our results. *Crow et al. (2015)* reported post-harvest herbicides could prevent up to 1,200 palmer amaranth seeds m$_{-1}$ from being incorporated into the soil. In our study, the effect of PHG, with or without post-emergence tribenuron-methyl, on the weed seed bank was lower than that in that study. This discrepancy is likely due to the weeds in our cropping system. Palmer amaranth has a potential to produce thousands of seeds in a season (*Crow et al., 2015*). Our results indicate that effective weed seed bank management in continuous monoculture winter wheat can be achieved by integrating weed control methods such as PHG, hand weeding, and post-emergence herbicide applications. The main factor driving this effect is likely disruption of the weed life cycle, similar to the impact of crop rotation and/or tillage (*Feledyn-Szewczyk et al., 2020*).

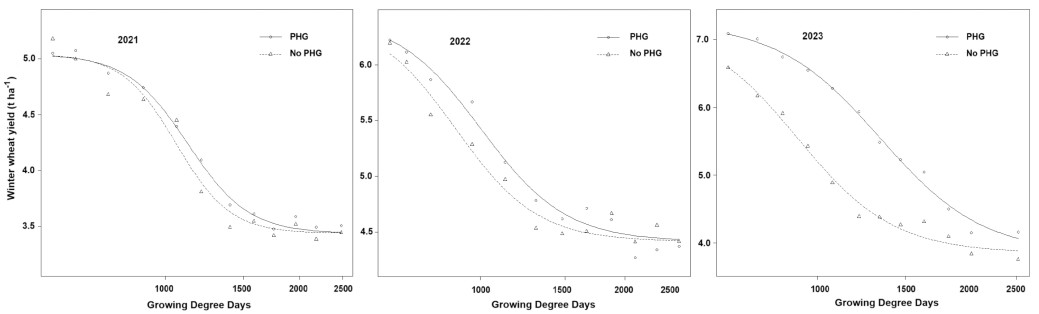

**Figure 4** Winter wheat yield (t ha$^{-1}$) in response to increasing duration of weed interference as represented by growing degree days (GDD) after wheat emergence for Post-harvest glyphosate (PHG) and No PHG applications at Kazova in 2021–2023.

## Impact of the PHG on the critical time for weed removal
### Wheat yield loss

Wheat yield losses were affected by the timing of weed removal, PHG, and year (Fig. 4). The wheat yield decreased with increasing duration of weed interference during 2021–2023. The grain yields in season-long weed-free plots with PHG were 5.127 t ha$^{-1}$ in 2021, 7.668 t ha$^{-1}$ in 2022, and 7.120 t ha$^{-1}$ in 2023, respectively. The wheat yield in the season-long weedy plots with PHG was 46.1, 75.5, and 71.2% lower than that in the season-long weed-free plots. Similarly, the yields in the season-long weed-free plots without PHG were 5.211 t ha$^{-1}$, 6.510 t ha$^{-1}$, and 6.635 t ha$^{-1}$, respectively, during the same period. The wheat yield in the season-long weedy plots without PHG was 51.3, 47.5, and 76.6% lower than that in the season-long weed-free plots. Notably, the smallest yield decrease was observed in 2022 compared with 2021 and 2023, which was due mainly to weed presence and reduced rainfall (38.93% lower than the average).

Wheat yield loss due to weeds is reported to be 3.0–34.4% in the USA, 2.9% in Canada, and 20–32% in Pakistan (*Flessner et al., 2021*; *Chhokar, Sharma & Sharma, 2012*). These rates were lower than our results, which was caused mainly by the specific weed society in our cropping system and rainfall. Under soil moisture deficiency such as in our study, competition between weeds and wheat is greater meaning reduced wheat yields when compared to favorable conditions (*i.e.,* high moisture) (*Ihsan, El-Nakhlawy & Ismail, 2015*). *Webster, Cardina & Loux (1998)* investigated the impact of post-harvest herbicide application (glyphosate+2,4-D) at three application times on corn yield and weed biomass in the corn field grown in the winter wheat-corn cropping system. They found that weed biomass was significantly reduced by post-harvest herbicide application without herbicide treatment, as the corn yield was 2.75 times higher than that of non-treated control, similar to our findings.

### Critical time for weed removal

The CTWR was calculated using the log–logistic model described based on a 5% acceptable yield loss by *Ritz et al. (2015)*. In the first year of the study, the CTWR began at nearly the same point in both the No PHG and the PHG treatments (Table 4, Fig. 4). Specifically,

**Table 4  Regression parameters (standard error) and estimation of CTWR (±SE) for No PHG and PHG applications at Kazova in 2021–2023.**

| Year | Treatment | B | C | D | $I_{50}$ | GDD | DAE |
|------|-----------|-----|------|------|----------|------|------|
| 2021 | No PHG | 7.2 (2.7) | 3.40 (0.07) | 5.02 (0.13) | 10.73 (0.50) | 711.9 (131.8) | 31.47 (7.1) |
|      | PHG | 6.8 (1.1) | 3.48 (0.04) | 5.05 (0.06) | 11.1 (0.26) | 718.5 (55.3) | 31.1 (2.8) |
| 2022 | No PHG | 4.5 (1.3) | 4.44 (0.06) | 6.85 (0.74) | 7.99 (1.24) | 416 (140.6) | 16.6 (3.0) |
|      | PHG | 4.3 (1.1) | 4.32 (0.09) | 6.47 (0.25) | 10.22 (0.63) | 516.5 (112.9) | 22.3 (2.8) |
| 2023 | No PHG | 4.2 (0.5) | 3.82 (0.14) | 7.25 (0.12) | 13.38 (0.33) | 465.6 (92.2) | 14.5 (2.2) |
|      | PHG | 4.4 (0.9) | 3.84 (0.10) | 7.30 (0.48) | 9.05 (0.59) | 661.2 (55.8) | 31.6 (2.8) |

**Notes.**

PHG, Post harvest glyphosate; B, Slope; C, lower limit; D, Upper limit; $I_{50}$, the GDD at the inflection point; GDD, growing degree days; DAE, Days after emergence.

the CTWR was initiated at 711.9 GDD in the conventional system and 718.5 GDD in the PHG treatment, showing minimal differences between the two treatments in terms of the onset of critical weed pressure. In contrast, the apparent effect of the PHG treatment on the CTWR was observed in the second year. In 2022, the CTWR started at 416 GDD, which corresponds to 16.6 DAE in the No PHG treatment; meanwhile, the CTWR was delayed until 516.5 GDD (22.3 DAE) in the PHG treatment. The PHG treatment delayed the CTWR from 465.6 GDD (14.5 DAE) to 661.2 GDD, equivalent to 31.6 DAE in 2023. This means a substantial delay of 17.1 days compared with the No PHG treatment.

Previous studies have shown that the CTWR varies according to numerous factors, including the growing season, crop species, agronomic practices, and environmental factors (*Contreras et al., 2022*). However, few studies about CTWR in wheat have been reported in the literature. The CTWR started 2-3 weeks after seed emergence or 28 or 30 days after sowing (*Agostinetto et al., 2008*; *Morsy et al., 2020*; *Chaudhary et al., 2008*). These times agree with our results, which varied from 16.6–31.47 DAE in No PHG treatment.

The relationship between weed abundance and CTWR has been reported in many crops (*Charles et al., 2019*; *Charles et al., 2020*; *Williams, Ransom & Thompson, 2007*). Nearly all the results are more or less the same as weed density decreases, and CTWR is delayed. To reduce weed density in non-herbicide resistant crops, total herbicides might be used at various times, such as pre-sowing, pre-emergence, or post-harvest. These treatments improve weed suppression throughout the stubble or early growth stages and alter the CTWR. Pre-emergence herbicide treatments delay the CTWR by 15–31 d in soybean (*Roncatto et al., 2023*), 3–21 d in corn (*Ulusoy et al., 2021*), 6–12 d in sunflower (*Knezevic et al., 2013*), 31 d in dry bean (*Beiermann et al., 2022*), and 25–32 d in popcorn (*Barnes et al., 2019*). The impact of PHG treatment on the CTWR in our study, 5.7–17.1 d, was in line with these studies.

## CONCLUSIONS

Post-harvest glyphosate (PHG) application resulted in a gradual delay in the critical time for weed removal in the second and third years. The 5.6-day delay in 2022 and the 17.1-day delay in 2023 indicated how PHG can control weed growth and prolong the time before weed control practices. This delay provided by the PHG may offer farmers a longer period

to manage weeds in winter wheat. During this time frame, wheats continue to grow without weed competition.

PHG is a powerful tool for controlling many weeds in winter wheat and has the potential to be a substitute for post-emergence herbicides. This approach is particularly important in terms of preventing herbicide drift when winter wheat is grown adjacent to sensitive crops such as sunflower, sugar beet, legumes, and vegetables. Moreover, if PHG is combined with a post-emergence herbicide or hand weeding, it significantly improves grain yield while reducing weed density and weed seed density in the soil compared with those under conventional treatment.

Integrating PHG into conventional weed control practices in winter wheat can be a good alternative for controlling herbicide-resistant weeds without any extra effort or cost. Additionally, PHG + post-emergence herbicide or hand-weeding can be a more efficient tool to manage the soil seed bank and decrease weed pressure in the next crop. PHG is also a promising approach for keeping moisture and nutrients in the soil for subsequent crops, especially in semi-arid and arid regions.

### Funding
This study was supported by the General Directorate of Agricultural Research and Policy (TAGEM), project code is TAGEM/BSAD/E/20/A2/P4/1803. The funders had no role in study design, data collection and analysis, decision to publish, or preparation of the manuscript.

### Grant Disclosures
The following grant information was disclosed by the authors:
General Directorate of Agricultural Research and Policy (TAGEM): TAGEM/BSAD/E/20/A2/P4/1803.

### Competing Interests
Ahmet Tansel Serim is an Academic Editor for PeerJ.

### Author Contributions
- Ahmet Tansel Serim conceived and designed the experiments, analyzed the data, prepared figures and/or tables, authored or reviewed drafts of the article, and approved the final draft.
- Ünal Asav performed the experiments, prepared figures and/or tables, authored or reviewed drafts of the article, and approved the final draft.
- Yalçın Kaya performed the experiments, prepared figures and/or tables, authored or reviewed drafts of the article, and approved the final draft.
- Bülent Başaran performed the experiments, prepared figures and/or tables, and approved the final draft.
- Eric L. Patterson analyzed the data, prepared figures and/or tables, authored or reviewed drafts of the article, and approved the final draft.

## Data Availability

Raw data has been uploaded as a Supplementary File.

## Supplemental Information

Supplemental information for this article can be found online at http://dx.doi.org/10.7717/peerj.19177#supplemental-information.

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
