# Peer review of "Evaluating the role of post-harvest glyphosate application in enhancing weed control in winter wheat"

_PeerJ, doi:10.7717/peerj.19177_

## Round 0.1 · original submission · Minor Revisions

Minor revisions are sufficient for your manuscript. Do not overlook the attached files.

·

Basic reporting

The article is based on field experiment.
Sufficient background information is provided with defined problem and objectives.
Appropriate methods and procedures were followed.
Results are nicely presented and discussed scientifically with supporting references.
Conclusion has practical utility.
Sufficient and appropriate references are cited.

Experimental design

Appropriate

Validity of the findings

It has practical utility.

Additional comments

No

Reviewer 2 ·

Basic reporting

Reviewer opinions and suggestions
- Soil persistence and effect durations of the herbicides used in the study should be added to the appropriate place in the text.
- It should be explained why post-harvest application is included in the study. Are there any problems in pre-planting applications? and what are the advantages of post-harvest application over pre-planting application? If some weeds emerge from post-harvest application to planting, what kind of solution is offered should be stated. Is there a licensed commercial preparation for post-harvest glyphosate application in the world? It should be added to the introduction section.
-In the Figure 1 D. Separation of weed seeds from. It should be stated more clearly where the seeds come from and from which application.
-Line 136-138. “The treatments were: T1: PHG + Post-emergence tribenuron-methyl, T2: PHG + Weedy check, T3: No PHG + Post-emergence tribenuron-methyl, T4: No PHG + Weedy check, PHG + Hand weeding (T5), and No PHG + Hand weeding (T6)”. please write application names in the same format for example T5:...
- In the description of Figure 2, “(T1: PHG+Post-em Tribenuron; T2: PHG+Weedy check; T3: No PHG+Post-em tribenuron; T4: No PHG+Weedy check, T5: PHG+Hand weeding, T6: No: PHG+Hand weeding)”. omit “:”
- It was seen as a deficiency that pre-planting and pre-emergence applications were also included among the applications and not compared with post-harvest applications.
- The total active ingredient used in the study is the "potassium salt" form of glyphosate and should be given as "Glyphosate potassium salt" throughout the text. If there is no objection, a trade name may be given, but it is not mandatory. Is it a licensed preparation?, how were the doses determined.
- What is the reason for using different doses?
- Replace the word "Turkey" with the word "Türkiye" in the text.
In Average weed seed density (Table 3), Treatment of PHG+post-em tribenuron-methyl and PHG+ weedy check. What do you think is the reason why the applications made in 2023 will be different from those made in 2021 and 2022 (in the same treatment). Explain and add to the text.
- What do you think is the reason why the applications made in 2023 will be different from those made in 2021 and 2022? Explain and add to the text.
- How was it calculated that PHG application effectively prevented yield loss by delaying CTWR and minimized the soil seed bank, i.e. what was it compared to? Comparison applicationis T4 or T5. Why was a character that had no application added?
- If there are weeds for which herbicides are not effective, please specify them separately for each application.
- Which weed species seeds were identified in the seed bank studies?
- Literature should be added to the method section titled "Impact of PHG on CTWR"
- In the method section in the titled "Impact of PHG on CTWR" PHG or PHG + Post-emergence tribenuron-methyl
- The manuscript text and literature section should be written according to the journal writing rules. The format is not appropriate.

Experimental design

It would have been more appropriate to include a character to which no application was made as a control in the study.

Validity of the findings

no comment

Additional comments

no comment

Reviewer 3 ·

Basic reporting

The article is well-written in English but some mistakes in technical uses existed. Additional references are needed in M&M section. The abstract is too long so it should be shortened. One or two introduction sentences explaining the problem are sufficient.
Please use the country name Türkiye or Turkey. in Line 21, it was Türkiye, and in line 62 was Turkey.

Experimental design

The experiment is well-designed. However,
-Weed removal timing is not appropriate for technical use. It should be replaced by weed control timing.
-It needs to be specified in the text whether winter wheat is bread wheat (Triticum aestivum L.) or durum wheat (Triticum durum L.).
-What is the name of the winter wheat variety?
-Please explain whether hand weeding is a practical method of weed management and it is traditionally used in wheat cultivation in Türkiye.
-In line 142, ...ears in a 1 m2 area .... I think "ears" is not appropriate. It must be "spike"
-In line 190, please give the reference for GDD.
-Please explain how the weed density was measured.

Validity of the findings

The findings are very interesting for readers and the results can be used in practice. It may also be helpful for researchers interested in exploring different approaches to herbicide use.

---

## Round 0.2 · Minor Revisions

Robert Winkler, the Section Editor, has commented and said:

"Before acceptance of the paper, I recommend some additional requests of the authors:

Please revise the equations. The capitalization in the formula should be congruent with the variable explanation. I.e., if d is given in the equation, it should be d in the explanation, not D. Try to be concise. E.g. in the Conclusions: "This research showed that .." may be deleted. Instead, you should spell out the acronym PHG in the first occurrence of the section to make the text more easily readable."

·

Basic reporting

The article is revised as per suggestion.

Experimental design

Appropriate

Validity of the findings

Findings are valid and practically useful

Additional comments

No

Reviewer 3 ·

Basic reporting

It is suitable for publication.

Experimental design

It is suitable for publication.

Validity of the findings

It is suitable for publication.

Additional comments

It is suitable for publication.

---

## Round 0.3 · accepted · Accept

The changes you make are adequate for accept to me for your manuscript.

The Secti0on Editor noted:

> For readability I suggest spelling out the acronyms in the Conclusions. Currently, the text is somehow cryptic: "PHG application resulted in a gradual delay in the CTWR in the second and third years." These changes can be performed in the proof.